# Characterization of an unusual carlavirus-like RNA from papaya (*Carica papaya*) lacking essential genes

Milenka Vera[1,2], Raúl Cifuentes[1], Ronan Keener[2], Neil Olszewski[3], Robert A. Alvarez-Quinto[2]*, Diego F. Quito-Avila[1,4]*

1 Facultad de Ciencias de la Vida, Escuela Superior Politécnica del Litoral, ESPOL, Km 30.5 Vía Perimetral, Campus Gustavo Galindo, Guayaquil, Ecuador, 2 Department of Plant Pathology, University of Minnesota, Saint Paul, Minnesota, United States of America, 3 Department of Plant Biology, University of Minnesota, Saint Paul, Minnesota, United States of America, 4 Centro de Investigaciones Biotecnológicas del Ecuador, CIBE, Escuela Superior Politécnica del Litoral, ESPOL, Km 30.5 Vía Perimetral, Campus Gustavo Galindo, Guayaquil, Ecuador

* alvar419@umn.edu (RAA-Q); dquito@espol.edu.ec (DFQ-A)

## Abstract

We report the characterization of a novel carlavirus-like RNA, provisionally named papaya defective virus 1 (PapDfV1), identified through high-throughput sequencing of papaya latex RNA. PapDfV1 contains two open reading frames (ORFs): ORF 1 encodes a 211.1 kDa replicase with 96% sequence identity to Zhejiang betaflexivirus 2 (ZhBV2), while ORF 2 exhibits a chimeric structure with regions homologous to two distinct ORFs of ZhBV2. Notably, PapDfV1 harbors a 2.1 kb deletion spanning the downstream portion of triple gene block (TGB) 1, the entirety of TGB2 and TGB3, capsid protein (CP), and the upstream region of the nucleic acid binding protein (NABP). Phylogenetic analysis of the replicase sequence grouped PapDfV1 with ZhBV2 and three Chinese isolates of cowpea mild mottle virus (CPMMV), two of which share the terminal regions 5'-GGAAAAACAGAA and CCTGGTTTT-3'. Mechanical inoculation and agroinfiltration failed to infect PapDfV1 in papaya, *Nicotiana benthamiana*, and common bean, whereas a ZhBV2 construct successfully infected soybean, common bean, and *N. occidentalis*. Detection of PapDfV1 in total and double-stranded RNA from leaf tissue of a 2-year-old papaya plant indicates that its replication and movement may depend on host proteins expressed in mature papaya plants or be facilitated by co-infection with unknown papaya viruses. These findings contribute to the growing body of knowledge on the existence and evolution of non-canonical virus-like RNA sequences lacking CP and other genes typically involved in virus movement.

which permits unrestricted use, distribution, and reproduction in any medium, provided the original author and source are credited.

**Data availability statement:** NCBI GenBank Acce. Number: OR253703 https://www.ncbi.nlm.nih.gov/nuccore/OR253703.

**Funding:** The author(s) received no specific funding for this work.

**Competing interests:** The authors have declared that no competing interests exist.

## Introduction

The genus *Carlavirus* (family *Betaflexiviridae*), comprises single-stranded positive-sense RNA viruses enclosed in non-enveloped filamentous virions of 600 to over 1,000 nm in length and a diameter of 12–15 nm. The monopartite genome of carlaviruses varies from 7.4 to 8.7 kilobases (kb) in length, featuring a 5' cap and a 3' polyadenylated (poly(A)) tail [1]. Genome organization of carlaviruses includes six open reading frames (ORFs). ORF 1 encodes the viral replicase, a 215–225 kilodalton (kDa) protein with methyltransferase, RNA-dependent-RNA-polymerase (RdRp), endopeptidase, and helicase (HEL) domains. The contiguously arranged, usually overlapping, ORFs 2, 3, and 4 encode three proteins of 25, 12, and 7 kDa, known as the triple gene block (TGB) proteins 1, 2, and 3 (TGBp1, TGBp2, and TGBp3), respectively. TGB proteins have complementary roles involved in cell-to-cell and long-distance virus movement [2]. ORF 5 codes for a 32–36 kDa capsid protein (CP) and ORF 6 codes for a small, 11–16 kDa, cysteine-rich nucleic acid binding protein (NABP) [3].

Currently, the International Committee on Taxonomy of Viruses (ICTV) recognizes over 60 carlavirus species, with several others still classified as tentative species (https://ictv.global/taxonomy accessed on December, 2024). Carlaviruses infect a wide range of plant hosts across various taxonomic groups causing symptoms that vary from mild to severe. These viruses are primarily transmitted in a non-persistent manner by aphids, although transmission via the whitefly *Bemisia tabaci* (Gennadius) has also been documented for cowpea mild mottle virus (CPMMV) [4].

Papaya (*Carica papaya* L.) is an important tropical fruit grown in several countries. Numerous viruses have been reported in papaya, some of which have long been implicated in economically important diseases [5]. Recently, three carlaviruses: CPMMV, papaya mottle-associated virus (PaMV), and papaya mild mottle-associated virus (PaMMV) were found in papaya plants affected by ringspot disease in Kenya [6,7]. PaMV and PaMMV are related more closely to cucumber vein-clearing virus, whereas CPMMV is a well characterized whitefly-transmitted carlavirus originally found in cowpea (*Vigna unguiculata*) and later reported in other hosts [8,9].

In this study, we report the discovery of a novel carlavirus-like RNA in papaya. While it shares high sequence identities with replicases from CPMMV and other carlaviruses, it is distinct in containing only one of the six canonical ORFs typically found in carlavirus genomes. Our findings contribute to the growing body of evidence on the evolution of non-canonical virus-like RNA sequences that lack genes traditionally considered essential, such as those found in umbra-like viruses, which contain only replicase-associated genes yet remain infectious by exploiting host factors [10].

## Materials and methods

### Ethics statement

This research does not contain any assays involving human participants or animals. Field work was carried out in strict accordance with the *Genetic Resource Access Permit* # MAE–DNB–CM–2018–0098 granted by the Department of Biodiversity of the Ecuadorean Ministry of the Environment.

## High-throughput sequencing and contig assembly

Five hundred µl of latex from a single unripen fresh papaya fruit was collected for RNA extraction. Total RNA was extracted using the PureLink™ RNA Mini Kit (Thermo Fisher Scientific, USA). After DNase digestion, the RNA was submitted for high-throughput sequencing (HTS) using Illumina NovaSeq 6000 system (2x 150 bp). A cDNA library was generated using Illumina TruSeq Stranded kit after removal of plant ribosomal RNA with Ribo-Zero Plant. Sequence assembly was done using tools implemented in Geneious Prime 2022–2023. Duplicate and low-quality reads were removed using Dedupe and BBDuk, respectively. Reads were subjected to *de novo* assembly using SPAdes v. 3.15.5 [11] followed by BLAST analysis against the non-redundant nucleotide and amino acid NCBI databases using the HPC resources from the Minnesota Supercomputing Institute (MSI). All contigs with > 95% nt identity (query coverage > 35%) with plant-related genes were not considered for further analysis.

## Virus detection

Virus detection was performed via reverse-transcription (RT)-PCR. Total RNA was extracted from 100 mg of young fully-developed leaf tissue or 500 µL of papaya fruit latex, following the protocol described by [12]. Additionally, double-stranded RNA (dsRNA) was extracted from 15 g of leaf tissue from the original plant stored at −80°C, using a phenol-based protocol as described by Dodds et al (1984) [13]. This protocol was modified by substituting cellulose powder medium-sized fibers (Sigma-Aldrich, Cat. No. C6288) for Whatman CF-11. Reverse transcription was conducted with SuperScript™ IV (Thermo Fisher Scientific, USA) and random primers, according to the manufacturer's instructions. For RT from dsRNA, a denaturation step was performed by heating the dsRNA at 98°C for 10 minutes in the presence of random primers, and 10% DMSO followed by immediate quenching on ice. For total RNA, a DNase treatment using the ezDNase enzyme (ThermoFisher Scientific, USA) was applied prior to the RT step. PCR was performed in 10 µL volumes using Platinum™ SuperFi II DNA Taq Polymerase (Thermo Fisher Scientific, USA) and detection primers listed in S1 Table.

## Confirmation of genomic sequence and Rapid Amplification of Complementary DNA Ends (RACE)

The newly assembled contig was confirmed using dsRNA as a template for RT-PCR, with primers designed to span the complete sequence in an overlapping manner. Reverse transcription-PCR was done as described above. Terminal regions were determined using a second-generation RACE kit (Roche, Germany) with two virus specific primers designed at each end. PCR-amplified products were cloned using a pGEMT-easy Kit (Promega, USA) and sequenced by the Sanger method. S1 Table lists all primers used in this study.

## Search of missing genes and helper viruses

Two complementary approaches, *in silico* and RT-PCR-based, were employed to investigate whether the missing genes were present as part of a distinct "normal" RNA form or as an independent RNA molecule. For the *in silico* approach, a non-deleted viral RNA sequence (GenBank Acc. No. MW897315), corresponding to the closest BLASTn match was utilized to map the total HTS reads. Mapping was performed using BBMap [14] as implemented in Geneious Prime 2023.

The RT-PCR-based approach involved 25 µL reactions using primers designed to span the missing genomic region (S1 Table). Platinum™ SuperFi II DNA Taq polymerase was used for amplification. The PCR cycling conditions were an initial denaturation step at 98°C for 5 minutes, followed by 40 cycles of 98°C for 1 minute, 55°C for 30 seconds, and 72°C for 3 minutes, with a final extension at 72°C for 10 minutes. In addition, primers specific to the missing region based on the closest BLASTn match (GenBank Acc. No. MW897315) were used for PCR. These primers were previously validated on a synthetic fragment of GenBank Acc. No. MW897315, which included the primer binding sites (Macrogen Inc., Seoul, South Korea).

To investigate the presence of CPMMV, a carlavirus recently reported in papaya, as a potential virus helper, primers CPMMV-F (CGATCCTGTCGAGATTGGTT) and CPMMV-R (AGAAGGCCCTCAAATCATCC) were used as recommended [15]. In addition, papaya-infecting potexviruses such as papaya mosaic virus (PapMV) and papaya virus X (PapVX), which could potentially act as helper viruses complementing the functions of the TGB and CP, were tested employing generic primers PapVX5 (CACCARCARGCNARRGATGA) and PapVX1RC (TCDGTGTTKGCRTCRAADGT), following the protocol described by Cabrera-Mederos et al (2022) [16].

## Sequence analysis

The complete ORF 1 of the newly assembled sequence, along with 110 homologues from CPMMV isolates, and additional papaya infecting carlaviruses including PaMV, PaMMV (6), and papaya chlorotic spot associated virus (PaCSaV), an unpublished carlavirus sequence record (GenBank Acc. No. MZ173553) were used for Multiple Sequence Alignments (MSAs) and phylogenetic analysis. MSAs were done at the nucleotide and amino acid level using MUSCLE [17] implemented in Geneious 2023. Phylogenetic analyses were done using the Maximum-Likelihood method with 1,000 bootstrap replications and a JTT+G+I+F model implemented in MEGA X [18].

## Mechanical inoculation

Papaya latex, previously confirmed positive for the virus by RT-PCR, was used to mechanically inoculate virus-free papaya, common bean (*Phaseolus vulgaris* L.), and *Nicotiana bethamiana* plants with the carlavirus-like RNA. Two approaches were employed for latex inoculation. In the first approach, the latex was mixed with 0.5 M phosphate buffer (pH 7.0) at a 1:1 (w/v) ratio and applied to the two youngest fully developed leaves of each test plant (*n* = 6 per host) using a soft sponge. The leaves were pre-dusted with silicon carbide (carborundum, Sigma, USA) to facilitate infection. In the second approach, 20 µL of pure latex was deposited into a 1-cm deep stem bevel cut made in papaya and common bean (*n* = 6 per host). The wounds were immediately sealed with parafilm after inoculation. Inoculated plants were tested for the presence of the virus at 30- and 60-days post-inoculation (dpi) using total RNA extraction, as described above.

## Viral constructs

Two viral constructs were assembled into the pLX-AS binary vector. The constructs, one corresponding to the carlavirus-like RNA sequence (6,071 nt) with the naturally-occurring deletion (pLX-AS_D) and the other (pLX-AS_MW) representing the complete genome sequence (8,181 nt) of the closest relative available in GenBank (Accession No. MW897315), were synthesized by Twist Bioscience (California, USA) and assembled using the NEBridge® Golden Gate assembly kit (New England Biolabs, USA) following the protocol described by [19].

Chemically synthesized DNA fragments for the assembly of each construct were flanked with convergent AarI restriction sites for directional cloning into AarI-digested pLX-AS between the CaMV 35S promoter and NOS terminator (S1 Fig). The ribozyme sequence (5'-GTCACCGGATGTGCTTTCCGGTCTGATGAGTCCGTGAGGACGAAAC-3') from tobacco ringspot virus satellite RNA (GenBank Accession No. M14879) was appended after a 13nt-long poly A tract at the 3' end of the genome. To facilitate ribozyme folding and processing, a 9-nt stretch of thymine was added downstream of the ribozyme sequence. The constructs were transformed into chemically competent TOP10 *Escherichia coli* cells and the sequence confirmed by restriction analysis (FastDigest, Thermo Fisher Scientific, USA) followed by full-length plasmid DNA sequencing using Oxford Nanopore technology (Plasmidsaurus, Oregon, USA).

## Inoculation of *Agrobacterium tumefaciens*

The selected clones were transformed into electrocompetent *Agrobacterium tumefaciens* strain AGL1 cells using a BTX Pulse Power Generator T100 as described [20]. Recombinant cells were selected with LB (1.0% Tryptone, 0.5% Yeast Extract and 0.5% NaCl) supplemented with carbenicillin (100 µg/ml) and kanamycin (50 µg/mL) and confirmed by colony

PCR using a set of a vector-specific and virus-specific primers (S1 Table). Plasmids from two colonies for each construct were extracted and used to clone back into TOP10 *E. coli* for full-length plasmid sequencing as described above. Once confirmed, a single agrobacterium colony for each construct was cultured in antibiotic-supplemented LB overnight ($OD_{600} \sim 1.0$). One mL of the culture was centrifuged at 2,600 g for 3 min and the resulting pellet was resuspended in 1 mL of sterile water. A young leaf was cut across the central vein, within the first third of the petiole, and a sterile toothpick was dipped into the water-resuspended bacteria and applied to the cut edge. Additionally, the bacteria-dipped toothpick was used to puncture the petiole of a separate leaf and the stem between the two nodes. This method (Olszewski unpublished, S1 Fig) was employed to inoculate three-to-six-week-old common bean (*Phaseolus vulgaris* L.) 'Verano' (n = 30), soybean (*Glycine max* L.) 'USDA-N7004' (n = 15) and 'LB18-58' (n = 15), *Nicotiana benthamiana* (n = 16) and *Nicotiana occidentalis* (n = 16), and five-to-eight-week-old papaya plants 'Maradol' (n = 16), 'Sunrise' (n = 16) and 'Passion red' (n = 8).

As a second approach, agrobacterium culture ($OD_{600} \sim 0.5$) was pelleted at 2,600 g for 10 minutes, and resuspended in an equal volume of infiltration buffer (MS medium supplemented with ascorbic acid 30 mg/L, 10 mM 3-(N-Morpholino) propanesulfonic acid, pH 5.5, and 200 µM acetosyringone). The infiltration culture was incubated at room temperature for 3 h under dark conditions prior to infiltration using a 1 mL needleless syringe applied to the abaxial surface of the two youngest fully developed leaves as recommended [21]. This method was applied to five-to-eight-week-old papaya plants 'Sunrise' (n = 12) and 'Passion red' (n = 8), and *N. benthamiana* (n = 13).

Plants were inoculated with *A. tumefaciens* containing either the pLX-AS_D or pLX-AS_MW construct. In addition, non-inoculated plants 'mock', or plants inoculated with *A. tumefaciens* containing the pLX-AS (empty vector), or the bacterial resuspension medium were used as negative controls. Plants were maintained under growth chamber or greenhouse conditions (14/10 hours (day/night) ~ 25°C) and monitored for symptoms. Virus testing was done on young non-inoculated leaves at 7, 15, 21 and 30 dpi as described above. RT-PCR amplicons were purified and verified by Sanger sequencing.

### Semi-purification of viral particles and transmission electron microscopy

Virus-positive plants obtained through agroinoculation were subjected to virus purification following the protocol described by [22]. Briefly, 2 g of fresh leaf tissue were ground with liquid nitrogen and homogenized with 4 ml of extraction buffer (0.02 M borate buffer, pH 9.5), and filtered using a cheesecloth. The extract was centrifuged for 15 min at 10,000 g using a Beckman JA-17 Fixed-Angle rotor and the supernatant was mixed with 0.5 volumes of chloroform in a shaker for 4 min and incubated at 4°C for 1h. The mixture was centrifuged for 15 min at 10,000 g and the aqueous phase was filtered through medium porosity P5 filter paper (Fisher Scientific, USA). Viral particles were sedimented by centrifugation at 75,000 g using a Beckman Type 50.2 Ti Fixed-Angle rotor for 1h and suspended in 1 mL of 0.01 M borate, pH 9.5. The partially purified viral suspensions were mounted onto carbon-coated formvar (1%) grids, stained with 2% phosphotungstic acid (PTA) pH 7.0 and visualized by transmission electron microscopy (TEM) using a JEOL JEM-1400 Plus microscope at the University of Minnesota Imaging Center. Particle length and diameter were verified with ImageJ software.

## Results

### Sequencing and BLAST

A total of 42.3 million reads were generated by HTS. Aside from hundreds of plant-related contigs identified by BLASTn, a single 6 kb contig was assembled from 2,337 reads (0.006% of total reads), with an average read depth of 57x. The contig contained two ORFs (ORF 1 and 2) of 5.6 and 0.3 kb in length, respectively. BLASTn analysis of ORF 1 showed homology with 96% identity, and 100% coverage, to the putative replicase of Zhejiang betaflexivirus 2 (ZhBV2), a carlavirus-like RNA identified in China (GenBank Acc. No. MW897315). Additional related homologues for ORF 1 corresponded to several isolates of CPMMV, with an average of 76% nucleotide (nt) identity. Analysis of ORF 2 revealed a chimeric sequence encompassing 180 nt with homology to the upstream region of ZhBV2 TGB 1, and 171 nt homologous to the downstream region of ZhBV2 NABP gene (Fig 1A).

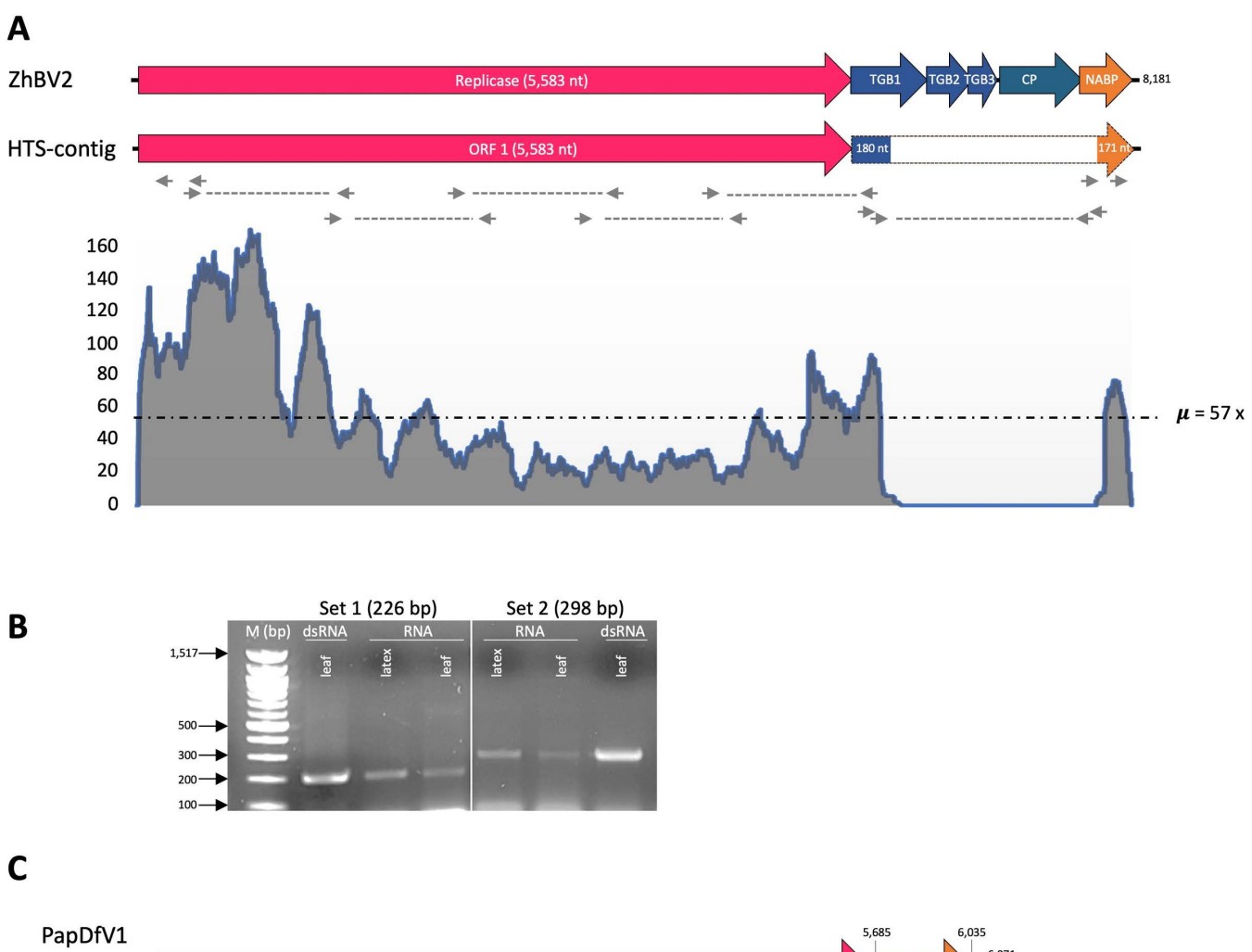

**Fig 1. Genome assembly, organization, and detection of papaya defective virus 1 (PapDfV1).** A) The organization of the initial contig assembled from high throughput sequencing (HTS) is shown alongside Zhejiang betaflexivirus 2 (ZhBV2), its closest relative. The canonical carlavirus genome structure of ZhBV2 is illustrated, featuring open reading frames (ORFs) for the replicase, triple gene block (TGB) 1–3, capsid protein (CP), and nucleic acid-binding protein (NABP). Notably, PapDfV1 exhibits a chimeric 351 nucleotide ORF 2 (represented as a dotted boxed arrow), containing sequences homologous to TGB 1 and NABP. Small arrows connected by dotted lines indicate primer sites used for genome re-amplification, while primers at the genome ends represent those used for RACE. Note that the deletion site was flanked by two primer sets. A read abundance graph displays the number of nucleotide reads mapped per site on the contig, illustrating sequencing coverage. B) Agarose gel electrophoresis showing two DNA fragments amplified by reverse-transcription (RT)-PCR. Both fragments correspond to the TGB1-NABP junction of PapDfV1. RT-PCR was done using two primer sets tested on total RNA and double-stranded RNA (dsRNA) extracted from leaf or latex tissues. The amplified products are displayed alongside a 100 bp DNA ladder (New England Biolabs, USA) used as a molecular size reference. C) The complete PapDfV1 genome sequence is depicted, including the coordinates of the ORFs and the molecular weight of the replicase encoded by ORF 1 and the hypothetical chimeric ORF 2.

## Virus detection and genome assembly

Two distinct primer sets, designed from TGB1 and NABP regions of the newly found carlavirus-like sequence, amplified fragments of 226 and 298 nt in length, respectively, from total RNA extracted from latex and leaf tissues, as well as from

dsRNA extracted from leaves (Fig 1B). Sequencing of overlapping RT-PCR products spanning the entire sequence, combined with RACE analysis, revealed that the complete sequence of the carlavirus-like RNA (GenBank Accession No. OR253703) is 6,071 nucleotides long, excluding the 3' poly(A) tail. Notably, sequencing of the amplified region spanning the TGB1 and NABP confirmed the chimeric nature of ORF 2 (Fig 1C). Terminal sequences (5'-GGAAAAACAGAA and CCTGGTTTT-3') are identical to those of two soybean isolates of CPMMV (GenBank Accession No. OK625819 and MT366555) but differ from those of ZhBV2, its closest relative. Additional *in silico* analyses failed to detect the missing genomic region encompassing the downstream portion of TGB1, the complete TGB2 and TGB3, the CP, and the upstream region of the NABP gene. No evidence of CPMMV or a potential helper potexvirus was found. These findings indicate a substantial deletion of approximately 2.1 kb spanning these genes (Fig 1A, C). This novel carlavirus-like RNA is provisionally named papaya defective virus 1 (PapDfV1).

## Phylogenetic analysis

The putative replicase encoded by PapDfV1 ORF 1 (nucleotide positions 74–5,656) is composed of 1,859 amino acids with a predicted molecular weight of 211.1 kDa. This protein shares 96% sequence identity with the replicase of ZhBV2 and 72–79% identity with those of CPMMV isolates. In contrast, its amino acid sequence identity with the replicases of papaya-associated carlaviruses (e.g., PMaV, PaMMV, and PaCSaV) was approximately 40%.

Accordingly, the phylogenetic analysis revealed two well-defined groups: a main cluster containing most CPMMV isolates and a group comprising PapDfV1, ZhBV2, along with three CPMMV isolates from China, including those with identical terminal sequences (Fig 2).

## Virus inoculation

Mechanical inoculation of virus-positive papaya latex onto leaves or stem cuts failed to transmit the virus to papaya, *N. benthamiana*, or common bean plants. For the agroinoculation, viral construct pLX-AS_MW resulted in successful virus infection in soybean ($n = 7/12$), common bean ($n = 12/12$), and *N. occidentalis* ($n = 5/5$) detected at 15 or 21 dpi. Conversely, agroinoculation with pLX-AS_D did not result in infected plants in any of the tested species, regardless of the agroinoculation method used. Carlavirus-like particles exhibiting a modal length ranging between 660–670 nm ($n = 10$) and 14.0 nm in diameter were visualized from agroinoculated soybean and common bean (Fig 3), but not in *N. occidentalis*. Agroinoculated plants did not exhibit discernible virus-like symptoms.

## Discussion

We present evidence of an unusual carlavirus-like RNA, designated PapDfV1, which exhibits a ~2.1 kb deletion encompassing the downstream region of TGB 1, the entirety of TGB 2, TGB 3, and CP, as well as the upstream portion of NABP. Although uncommon, such deletion is not unprecedented. A significant deletion affecting nearly the entire TGB 1-encoding ORF was previously reported in an isolate of the carlavirus butterbur mosaic virus (ButMV) from *Veronica* sp., an ornamental species [23]. Interestingly, the unusual ButMV variant (ButMV-B) was found in co-infections with its normal counterpart (ButMV-A) and helenium virus S (HelVS), another carlavirus, suggesting functional complementation of the missing TGB 1 gene of ButMV-B [23].

In our study, multiple approaches failed to identify a potential complementing virus—whether a "native" form of PapDfV1 or a related virus with a similar genomic structure, such as potexviruses. To investigate the infectious nature of PapDfV1 in absence of a helper virus, we developed a construct of the virus to agroinoculate virus-free plants. However, this construct failed to infect any of the tested plants, including papaya. In contrast, a construct of ZhBV2, PapDfV1's closest relative (96% nt identity in the replicase) –used to validate the agroinoculation methodology– successfully infected soybean, common bean, and *N. occidentalis*, but not papaya.

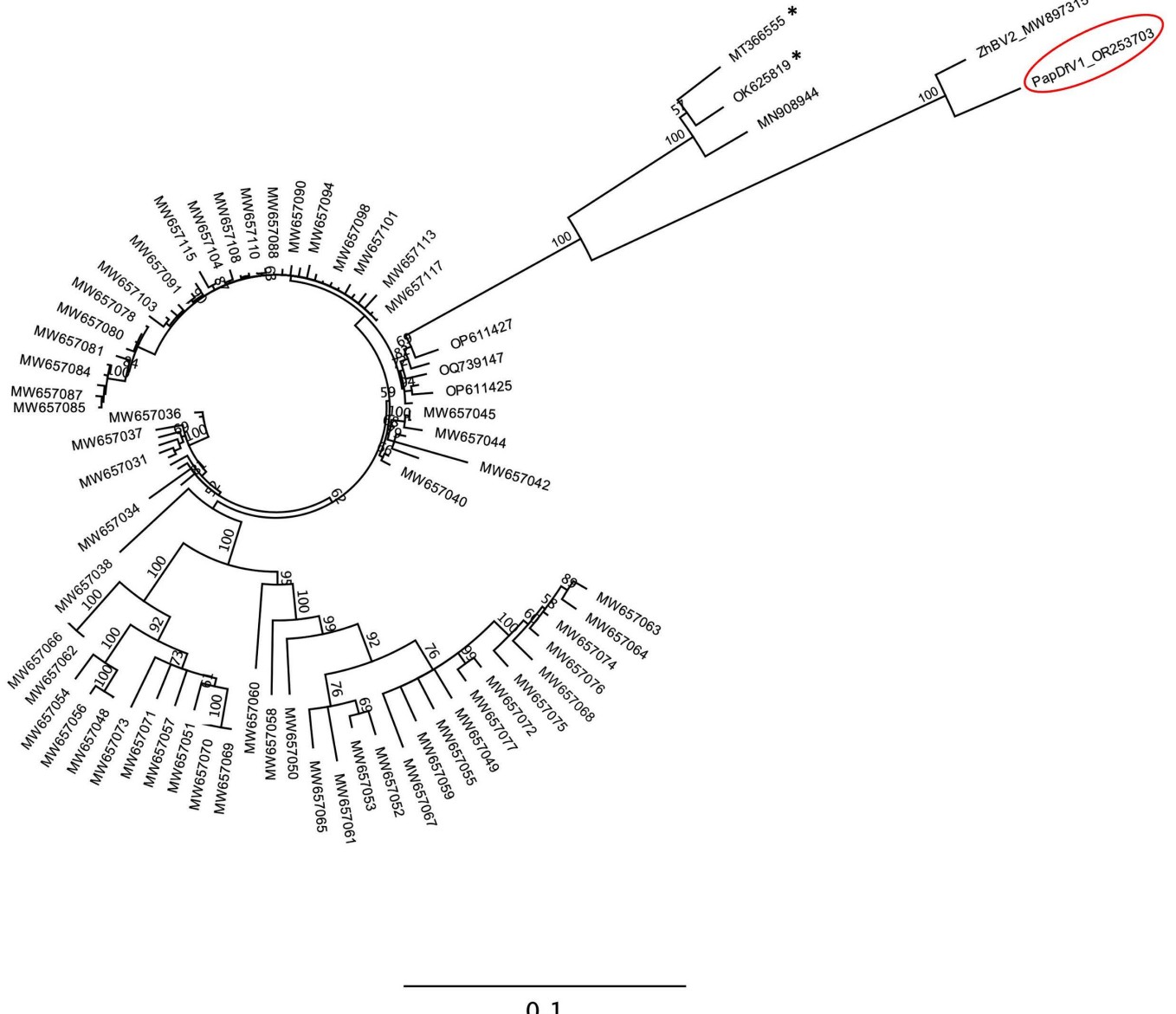

**Fig 2. Phylogenetic analysis of papaya defective virus 1 (PapDfV1).** The phylogenetic tree was constructed based on the complete amino acid sequence of the PapDfV1 replicase, along with 100 homologous sequences retrieved from GenBank, including the sequence of Zhejiang betaflexivirus 2 (ZhBV2) and several isolates of cowpea mild mottle virus (CPMMV). Taxa corresponding to CPMMV isolates are labeled with their respective GenBank accession numbers. CPMMV isolates marked with an asterisk (*) share identical terminal sequences with the genome of PapDfV1 at both genomic termini, whereas isolates without the asterisk lack sequence conservation at one or both termini. The analysis was performed using the Maximum-Likelihood method, employing the JTT+G+I+F substitution model, with 1,000 bootstrap replications (bootstrap values are indicated at each node). The tree was generated using MEGA X software.

Despite the lack of experimental evidence for systemic infection and movement of PapDfV1, its infection under natural conditions was supported by its detection in total RNA and dsRNA both extracted from leaf tissue. Notably, RT-PCR amplification of the fragment spanning the deleted region was prominent from dsRNA, indicating active replication of viral RNA not only in latex but also in leaf tissues. This raises the unresolved question of how PapDfV1 achieves cell-to-cell

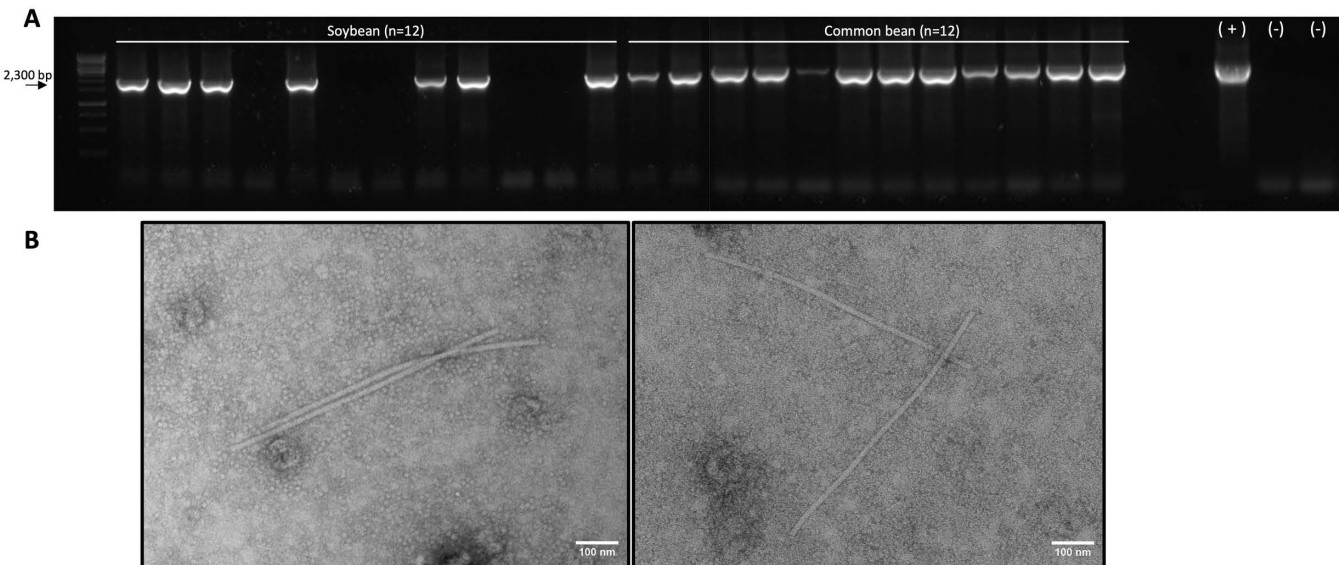

**Fig 3. Virus construct infectivity.** A) Agarose gel electroforesis of reverse-transcription (RT)-PCR amplified fragments flanking a 2,300 bp region spanning the TGB and NAB genes. Results correspond to soybean and common bean plants agroinoculated with pLX-AS_MW and tested at 30 days post-inoculation (dpi). The amplified products are displayed alongside a 1 kb plus DNA ladder (Thermo Scientific, USA) used as a molecular size reference B) Filamentous carlavirus-like particles visualized via transmission electron microscopy in semi-purified virus preparations from soybean (left) and common bean (right) plants following agroinoculation with the pLX-AS_MW construct.

and systemic movement in papaya. Since PapDfV1 was originally detected in a two-year-old papaya plant during the production stage, it is reasonable to speculate that its movement and replication may depend on host proteins. Such proteins may be expressed specifically during the mature stage but absent in younger plants (e.g., at eight weeks). Several host proteins interact with viral movement proteins (MPs) during intra- and intercellular and long-distance movement [24]. It has been demonstrated that viruses without essential movement genes such as umbra-like viruses and viroids use host proteins for systemic movement [25]. One such protein is phloem protein 2 (PP2), which exhibits RNA-binding activity and plays a role in plant defense as well as responses to both abiotic and biotic stresses [26]. Although PP2 is found in high concentration in the phloem exudate of *Cucurbitaceae*, its expression is host-dependent and influenced by environmental conditions. In a recent study by Ying et al (2024) [27], the citrus yellow vein-associated virus 1 (CY1), an umbra-like virus encoding only replication proteins, was shown to interact with the host PP2 to form a high molecular weight complex with sap proteins. This complex conferred protection against RNases and facilitated the virus's systemic spread. These findings align with previous studies implicating PP2 in the systemic movement of circular RNAs from viroids [28] and facilitating plant-to-plant transmission of cucurbit aphid-borne yellows luteovirus [29].

In addition, studies on trans-complementation of virus movement have shown that heterologous, unrelated viral MPs can complement the cell-to-cell spread of movement-deficient plant viruses or even viroid genomes. This is attributed to two conserved functions of many MPs: *i*) their ability to bind RNA and *ii*) their capacity to increase the size exclusion limit (SEL) of plasmodesmata (PD) [30]. Consequently, it is plausible to suggest that the co-infection with other papaya viruses present during sampling, such as the potyvirus papaya ringspot virus (PRSV) or the cytorhabdovirus papaya virus E (PpVE), both of which encode MPs, could potentially facilitate PapDfV1's systemic infection. Future studies should investigate these possibilities, as the defective nature of PapDfV1 might reveal novel functional complementarities between unrelated viruses or, alternatively, expand our understanding of host proteins exploited by infectious RNAs in plants.

Defective RNAs, including defective interfering RNAs (DIs) that inhibit the replication of their native viral forms [31], often arise from errors during viral replication and are characterized by the presence of hairpin structures, AU-rich elements, and repetitive sequences, which may predispose viral polymerases to template switching—a key mechanism in the generation of defective RNAs [32]. Interestingly, ButMV-B, a distantly related carlavirus, exhibits a similar deletion site, though to a lesser extent, suggesting a conserved genomic element associated with these deletions. Comparative studies with other defective viral RNAs may shed light on the genomic signals and mechanisms underlying the generation of these genomes, as well as their broader implications for viral pathogenesis, persistence, and co-evolution with hosts.

In summary, PapDfV1 represents a unique example of a defective carlavirus-like RNA with a substantial genomic deletion, raising intriguing questions about its origin, replication, and systemic movement in papaya. The findings highlight the importance of investigating both viral and host factors involved in facilitating the replication and persistence of such defective RNAs under natural conditions. The lack of evidence for PapDfV1 autonomous infectivity highlights the potential reliance on unknown helper viruses or host-specific factors. Further investigation of PapDfV1, including its transcomplementation with co-infecting viruses and interactions with host machinery, may uncover novel strategies used by defective RNAs to bypass functional limitations.

## Supporting information

**S1 Table. List of primer sequences used during this study.** Abbreviations: papaya defective virus 1 (PapDfV1), Zhejiang betaflexivirus 2 (ZhBV2), viral construct of papaya defective virus 1 (pLX-AS_D), viral construct of Zhejiang betaflexivirus 2 (pLX-AS_MW), Triple Gene Block 1 (TGB1), and Nucleic Acid Binding Protein (NABP). The expected size in nucleotides (nt) of each fragment is indicated. All primers were designed based on the sequences available in GenBank: OR253703 for PapDfV1, MW897315 for ZhBV2, and MW281334 for the cloning vector pLX-AS. Virus specific primers used for Rapid Amplification of cDNA Ends (RACE) were used with the anchored-oligodT primer provided in 5′/3′ RACE Kit, 2nd Generation (Roche, Germany).
(DOCX)

**S1 Fig. Construct design and agroinoculation methodology.** A) Schematic representation of the two viral constructs, pLX-AS_MW and pLX-AS_D, designed and tested in this study. Colored arrowed boxes denote open reading frames (ORFs), while black arrowed boxes denote left (LB) or right borders (RB) of the pLX binary vector. Sites corresponding to the 35S promoter from cauliflower mosaic virus (CaMV), the Ribozyme (RiboZ), the NOS terminator, and AarI restriction sites are indicated. B) Agroinoculation procedure showing the basic steps applied in this study. A steril toothpick was dipped into the bacterial suspension and was used to puncture the stem, or wet the edge of a cut leaf or a petiole.
(PDF)

## Acknowledgments

The authors thank Gail Celio at the University of Minnesota Imaging Center, for assistance during visualization of virus particles. We also appreciate Drs Fabio Pasin (CIB – CSIC) and Eduardo Sánchez (CIBE-ESPOL) for their valuable insights and recommendations on construct assembly and transformation methodologies, respectively, and Juan F. Cornejo-Franco for assisting us with the dsRNA extraction.

## Author contributions

**Conceptualization:** Diego F. Quito-Avila.

**Formal analysis:** Milenka Vera, Raul Cifuentes, Robert A. Alvarez-Quinto, Diego F. Quito-Avila.

**Funding acquisition:** Robert A. Alvarez-Quinto, Diego F. Quito-Avila.

**Investigation:** Milenka Vera, Raul Cifuentes, Ronan Keener, Neil Olszewski, Diego F. Quito-Avila.

**Methodology:** Milenka Vera, Raul Cifuentes, Ronan Keener, Neil Olszewski, Robert A. Alvarez-Quinto, Diego F. Quito-Avila.

**Supervision:** Neil Olszewski, Robert A. Alvarez-Quinto, Diego F. Quito-Avila.

**Writing – original draft:** Diego F. Quito-Avila.

**Writing – review & editing:** Milenka Vera, Ronan Keener, Neil Olszewski, Robert A. Alvarez-Quinto, Diego F. Quito-Avila.

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
