## [Decision Letter · Decision Letter 0]

23 May 2025

Dear Dr. Quito-Avila,

Thank you for submitting your manuscript to PLOS ONE. After careful consideration, we feel that it has merit but does not fully meet PLOS ONE’s publication criteria as it currently stands. Therefore, we invite you to submit a revised version of the manuscript that addresses the points raised during the review process.

We look forward to receiving your revised manuscript.

Kind regards,

Abozar Ghorbani, Ph.D

Academic Editor

PLOS ONE

Journal Requirements:

2. Please note that your Data Availability Statement is currently missing [the repository name and/or the DOI/accession number of each dataset OR a direct link to access each database]. If your manuscript is accepted for publication, you will be asked to provide these details on a very short timeline. We therefore suggest that you provide this information now, though we will not hold up the peer review process if you are unable.

3. Please amend the manuscript submission data (via Edit Submission) to include author Neil Olsewski.

4. Please amend your authorship list in your manuscript file to include author Neil Olszewski.

Reviewers' comments:

Reviewer's Responses to Questions

**Comments to the Author**

1. Is the manuscript technically sound, and do the data support the conclusions?

Reviewer #1: Partly

Reviewer #2: Yes

2. Has the statistical analysis been performed appropriately and rigorously?

Reviewer #1: Yes

Reviewer #2: Yes

3. Have the authors made all data underlying the findings in their manuscript fully available?

Reviewer #1: Yes

Reviewer #2: Yes

4. Is the manuscript presented in an intelligible fashion and written in standard English?

Reviewer #1: Yes

Reviewer #2: Yes

Reviewer #1: The manuscript entitled 'Characterization of an unusual carlavirus-like RNA from papaya (Carica papaya) lacking essential genes' represents the presence of a new carlavirus-like RNA elements in papaya latex. The authors have meticulously performed the experiments to justify the manuscript; but few points have to be addressed before publishing this manuscript:

1. Authors sampled only papaya latex, what about different plant parts or tissues?

2. The authors should check/confirmed the presence of CPMMV in the papaya plants harbouring the unusual carlavirus-like RNA as the helper virus; as both having close similarity with indentical terminal seqiences as written in Figure 2.

3. It is suggested to conduct a complementaion study by co-inoculating CPMMV or other carlavirues (PaMV, PaMMV) infecting papaya along with agroinoculation of pLX-AS_D to prove the infectivity of carlavirus-like RNA (PapDfV1) in papaya.

4. The quality of figures in the manuscript must improved.

5. Some minor typo-graphical errors (in attached files) must be corrected.

Reviewer #2: 1. In line 1, the title of the article is clear, attractive, and descriptive, and reflects the overall structure and content of the article

2. This article is written at a good level and the different sections for example introduction, abstract, discussion is readability and having sufficient organization.

3. In the introduction, the importance and position of the research are explained in an acceptable format.

4. In the results section, the data related to the phylogenetic analysis are explained in a suitable manner.

5. In line 38, add a short section on the importance of the discovery of PapDfV1 or other defective viruses in the Abstract and Introduction sections for understanding the evolution of plant viruses

6. In line 66, in the introduction section, provide a reference on possible mechanisms of compensation for deleted genes in carlaviruses or other viruses.

7. In line 226, in the results section, it is better to present a schematic diagram of the genomic structure of PapDfV1 compared to typical carlaviruses. To better understand their differences in terms of genomic structure, based on the figure

8. In line 373, in the conclusion section, further emphasis is placed on the importance of the findings of this research for the direction of future studies.

9. There are a number of sentences in the long text. Please write them in shorter sentences.

10. In line 373, the conclusion title should be written

11. The main headings, including introduction, materials and methods, etc., should be numbered respectively.

12. Please write references to format of journal

13. In total, this article contains a correct writing structure and very desirable content, and with some minor revisions and corrections, it is acceptable.

**Do you want your identity to be public for this peer review?** For information about this choice, including consent withdrawal, please see our Privacy Policy

Reviewer #1: No

Reviewer #2: **Yes: ** Farideh Farahbakhsh

Plant Protection Department, Agricultural Research, Education and Extension Organization (AREEO), Darab, Fars, Iran

---

## [Author Response · Author response to Decision Letter 1]

2 Jul 2025

Response to Reviewer’s Comments

We sincerely acknowledge the time and effort that both reviewers and the editor have dedicated to evaluating our manuscript. We are confident that each comment and suggestion was intended to enhance the quality and readability of our work, and we greatly appreciate the constructive feedback provided.

Below, please find our detailed responses to each of the reviewers' comments and suggestions. As part of our rebuttal, we have included a tracked-changes version of the manuscript for verification. Please note that we have carefully considered and addressed all comments, both those included in the email body and those provided in the attached documents.

Reviewer #1: The manuscript entitled 'Characterization of an unusual carlavirus-like RNA from papaya (Carica papaya) lacking essential genes' represents the presence of a new carlavirus-like RNA elements in papaya latex. The authors have meticulously performed the experiments to justify the manuscript; but few points have to be addressed before publishing this manuscript:

1. Authors sampled only papaya latex, what about different plant parts or tissues?

Response:

We appreciate this observation. The viral sequence was detected from total RNA extracted from both leaf and latex tissues, and additionally from dsRNA extracted from leaf tissue. We acknowledge that this detail was not explicitly stated in the Results section.

However, Figure 1C (in its current version) shows detection using two distinct primer sets from total RNA (from latex and leaf) and from dsRNA (from leaves). Furthermore, the Discussion does address this point:

“Despite the lack of experimental evidence for systemic infection and movement of PapDfV1, its infection under natural conditions was supported by its detection in total RNA and dsRNA, a replicative intermediate, both extracted from leaf tissue. Notably, RT-PCR amplification of the fragment spanning the deleted region was prominent from dsRNA, indicating active replication of viral RNA not only in latex but also in leaf tissues.”

Nevertheless, to directly address the reviewer’s observation, we have added a sentence in the Results section referring explicitly to the detection of the virus in the tested tissues. We have also slightly modified the subheading from “Virus genome assembly” to “Virus detection and genome assembly.” In this revised section, we now begin by describing the detection of the virus in both latex and leaf tissues.

To align with this change in narrative, we have also swapped the order of panels B and C in Figure 1.

We believe these adjustments, though minor, significantly enhance the clarity and readability of the manuscript.

2. The authors should check/confirmed the presence of CPMMV in the papaya plants harbouring the unusual carlavirus-like RNA as the helper virus; as both having close similarity with indentical terminal seqiences as written in Figure 2.

Response:

We appreciate the reviewer’s suggestion. We did test for the presence of CPMMV using CPMMV-specific primers; however, no amplification was obtained. Accordingly, we have added this information to the Methods section (see "Search for missing genes and helper viruses"), including the primer sequences and relevant citation. This has also been noted in the Results section (see the last paragraph of "Virus detection and genome assembly").

3. It is suggested to conduct a complementaion study by co-inoculating CPMMV or other carlavirues (PaMV, PaMMV) infecting papaya along with agroinoculation of pLX-AS_D to prove the infectivity of carlavirus-like RNA (PapDfV1) in papaya.

Response:

We acknowledge the validity and relevance of this suggestion.We have addressed this possibility as a direction for future studies in the Discussion section.

4. The quality of figures in the manuscript must improved.

Response:

We appreciate the reviewer’s comment. The reduced quality of figures in the current manuscript is due to the PDF conversion process. We assure that the final submitted figures will meet the high-quality standards required by the journal.

5. Some minor typo-graphical errors (in attached files) must be corrected.

Response:

Thank you. We have accepted all those typos diligently caught by the reviewer.

Reviewer #2:

In line 1, the title of the article is clear, attractive, and descriptive, and reflects the overall structure and content of the article

Response:

Thank you.

2. This article is written at a good level and the different sections for example introduction, abstract, discussion is readability and having sufficient organization.

Response:

Thank you.

3. In the introduction, the importance and position of the research are explained in an acceptable format.

Response:

Thank you.

4. In the results section, the data related to the phylogenetic analysis are explained in a suitable manner.

Response:

Thank you.

5. In line 38, add a short section on the importance of the discovery of PapDfV1 or other defective viruses in the Abstract and Introduction sections for understanding the evolution of plant viruses

Response:

Thank you. We have added the information suggested both at the end of the abstract and the introduction section.

6. In line 66, in the introduction section, provide a reference on possible mechanisms of compensation for deleted genes in carlaviruses or other viruses.

Response:

We respectfully disagree with this suggestion. The sentence referenced by the reviewer does not mention defective carlaviruses or the loss of viral genes, and thus, it would not be appropriate to cite compensation mechanisms in that specific context. However, we do address the concept of defective carlaviruses in the Discussion section, where we reference relevant literature and explain how previously reported events relate to our findings. We appreciate the reviewer’s attention to detail and thank them for their understanding.

7. In line 226, in the results section, it is better to present a schematic diagram of the genomic structure of PapDfV1 compared to typical carlaviruses. To better understand their differences in terms of genomic structure, based on the figure

Response:

We appreciate the reviewer’s suggestion. However, the genome organization of ZhBV2, the closest known relative of PapDfV1, is already presented in Figure 1 and represents the typical carlavirus genomic structure. As indicated in the figure legend, “The canonical carlavirus genome structure of ZhBV2 is illustrated.” Therefore, we believe that including an additional, separate schematic of a typical carlavirus genome would be redundant.

8. In line 373, in the conclusion section, further emphasis is placed on the importance of the findings of this research for the direction of future studies.

Response:

We appreciate the reviewer’s comment. We believe that the concluding paragraph already emphasizes the significance of our findings and their implications for future research. This emphasis is consistent with the context provided in both the Introduction and the Abstract, where we also highlight the relevance and potential of this study to guide further investigations.

9. There are a number of sentences in the long text. Please write them in shorter sentences.

Response:

We have shortened a few long sentences in the discussion section. We appreciate the detailed attention to this matter.

10. In line 373, the conclusion title should be written

Response:

We thank the reviewer for this observation. According to the PLOS ONE submission guidelines, including a titled "Conclusions" section is optional. As such, we have chosen to retain the current structure of the manuscript. Thank you for your understanding.

(see guidelines here: https://journals.plos.org/plosone/s/submission-guidelines).

11. The main headings, including introduction, materials and methods, etc., should be numbered respectively.

Response:

We thank the reviewer for the suggestion. However, PLOS ONE does not require numbering of main headings such as Introduction, Materials and Methods, etc. This is confirmed in the journal's formatting guidelines and the official manuscript template, which can be found here: https://journals.plos.org/plosone/s/submission-guidelines#loc-style-and-format:~:text=Download%20sample%20manuscript%20body%20(PDF)

12. Please write references to format of journal

Response:

Thank you. Yes. We have changed to the “Vancuver Style” required by the journal.

13. In total, this article contains a correct writing structure and very desirable content, and with some minor revisions and corrections, it is acceptable.

Response:

Thank you!

---

## [Decision Letter · Decision Letter 1]

21 Jul 2025

Characterization of an unusual carlavirus-like RNA from papaya (Carica papaya) lacking essential genes

PONE-D-25-08143R1

Dear Dr. Quito-Avila,

We’re pleased to inform you that your manuscript has been judged scientifically suitable for publication and will be formally accepted for publication once it meets all outstanding technical requirements.

Kind regards,

Abozar Ghorbani, Ph.D

Academic Editor

PLOS ONE

Additional Editor Comments (optional):

Reviewers' comments:

Reviewer's Responses to Questions

**Comments to the Author**

Reviewer #1: All comments have been addressed

Reviewer #2: All comments have been addressed

2. Is the manuscript technically sound, and do the data support the conclusions?

Reviewer #1: Yes

Reviewer #2: Yes

3. Has the statistical analysis been performed appropriately and rigorously?

Reviewer #1: Yes

Reviewer #2: Yes

4. Have the authors made all data underlying the findings in their manuscript fully available?

Reviewer #1: Yes

Reviewer #2: Yes

5. Is the manuscript presented in an intelligible fashion and written in standard English?

Reviewer #1: Yes

Reviewer #2: Yes

Reviewer #1: The authors have addressed all the question asked; this can be accepted for publication in the modified form.

Reviewer #2: Dear Editor,

All comments have been answered carefully and clearly. In my opinion, the article is eligible for acceptance in this journal.

All the best.

**Do you want your identity to be public for this peer review?** For information about this choice, including consent withdrawal, please see our Privacy Policy

Reviewer #1: No

Reviewer #2: **Yes: ** Farideh Farahbakhsh, Assistance Professor, Plant Pathology Department, Fars agricultural and natural resources, AREEO, Darab, Iran.

---

## [Editor Report · Acceptance letter]

PONE-D-25-08143R1

PLOS ONE

Dear Dr. Quito-Avila,

I'm pleased to inform you that your manuscript has been deemed suitable for publication in PLOS ONE. Congratulations! Your manuscript is now being handed over to our production team.

Kind regards,

on behalf of

Dr. Abozar Ghorbani

Academic Editor

PLOS ONE